# RNA-dependent RNA targeting by CRISPR-Cas9

**Steven C Strutt[1], Rachel M Torrez[1†], Emine Kaya[1‡], Oscar A Negrete[2], Jennifer A Doudna[1,3,4,5,6]\***

[1]Department of Molecular and Cell Biology, University of California, Berkeley, United States; [2]Sandia National Laboratories, Biotechnology and Bioengineering Department, Livermore, United States; [3]Howard Hughes Medical Institute, Maryland, United States; [4]Department of Chemistry, University of California, Berkeley, United States; [5]Innovative Genomics Institute, University of California, Berkeley, United States; [6]MBIB Division, Lawrence Berkeley National Laboratory, Berkeley, United States

**Abstract** Double-stranded DNA (dsDNA) binding and cleavage by Cas9 is a hallmark of type II CRISPR-Cas bacterial adaptive immunity. All known Cas9 enzymes are thought to recognize DNA exclusively as a natural substrate, providing protection against DNA phage and plasmids. Here, we show that Cas9 enzymes from both subtypes II-A and II-C can recognize and cleave single-stranded RNA (ssRNA) by an RNA-guided mechanism that is independent of a protospacer-adjacent motif (PAM) sequence in the target RNA. RNA-guided RNA cleavage is programmable and site-specific, and we find that this activity can be exploited to reduce infection by single-stranded RNA phage in vivo. We also demonstrate that Cas9 can direct PAM-independent repression of gene expression in bacteria. These results indicate that a subset of Cas9 enzymes have the ability to act on both DNA and RNA target sequences, and suggest the potential for use in programmable RNA targeting applications.
DOI: https://doi.org/10.7554/eLife.32724.001

**\*For correspondence:**
doudna@berkeley.edu

**Present address:** [†]Department of Medicinal Chemistry, University of Michigan, Michigan, United States; [‡]Global Blood Therapeutics, South San Francisco, United States

## Introduction

Prokaryotic clustered regularly interspaced short palindromic repeat (CRISPR)-CRISPR-associated (Cas) systems provide immunity against plasmids and bacteriophage by using foreign DNA stored as CRISPR spacer sequences together with Cas nucleases to stop infection (*Wright et al., 2016*; *Mohanraju et al., 2016*). One such nuclease, Cas9 of the type II systems, employs a CRISPR RNA (crRNA) and a trans-activating crRNA (tracrRNA) to target spacer-complementary regions (protospacers) on the foreign genetic element to guide double-stranded DNA cleavage (*Jinek et al., 2012*). A protospacer adjacent motif (PAM) must also be present for the Cas9-RNA complex to bind and cleave DNA (*Jinek et al., 2012*; *Gasiunas et al., 2012*; *Anders et al., 2014*; *Szczelkun et al., 2014*). Combining the crRNA and tracrRNA into a chimeric, single-guide RNA (sgRNA) simplified the system for widespread adoption as a versatile genome editing technology (*Jinek et al., 2012*).

To date, both genetic and biochemical data support the conclusion that *in vivo*, Cas9 is exclusively a DNA-targeting enzyme. Nonetheless, multiple studies have harnessed Cas9 for RNA targeting under specific circumstances. For example, the *S. pyogenes* Cas9 (SpyCas9) can be supplied with a short DNA oligo containing the PAM sequence (a PAMmer) to induce single-stranded RNA (ssRNA) binding and cutting (*O'Connell et al., 2014*; *Nelles et al., 2016*). More recently, it was demonstrated that SpyCas9 could be used to target repetitive RNAs and repress translation in certain mRNAs in the absence of a PAMmer (*Liu et al., 2016*; *Batra et al., 2017*). A different Cas9 homolog from *Francisella novicida* (FnoCas9) has been implicated in degradation of a specific mRNA

**eLife digest** Similar to humans, bacteria use an immune system known as the CRISPR-Cas system to protect themselves against invading pathogens such as viruses. CRISPRs are specialized stretches of DNA that guide Cas9 to the right location, while Cas9 proteins act like scissors that can cut foreign DNA.

When a virus infects a bacterium, the bacterium steals a piece of DNA from the virus and stores it in its CRISPR region. The bacterium then produces a small RNA template that matches the stolen DNA of the virus and adds a specialized protein to it. When the virus infects the cell again, the protein-RNA complex can recognize the virus and stop the infection.

Researchers have successfully adapted this system as a gene-editing tool to target and modify specific DNA sequences in different organisms. Cas9 can target and cut DNA, but until now, it was not clear whether this protein could also efficiently target RNA – the 'genetic middleman' between DNA and proteins. RNA is essential to make proteins, and being able to target RNA would allow researchers to answer many important questions about RNA biology.

To investigate this further, Strutt et al. used three different subtypes of Cas9 proteins and small RNA sequences in a test tube. The results showed that two of the protein subtypes could target RNA efficiently, and one of which was able to target any RNA sequence. Strutt et al. then used one Cas9 to target specific RNA sequences in bacteria and were able to reduce the amount of protein made from that gene. Moreover, the Cas9 protein helped to protect the bacteria against an RNA virus.

This work lays the foundation for using this Cas9 protein as a tool for researchers to study RNA in cells. A next step will be to test if Cas9 can cut RNA in human cells. If this works, it could allow direct targeting of RNA viruses, such as West Nile and Dengue, to stop them from infecting human cells.

DOI: https://doi.org/10.7554/eLife.32724.002

but through a mechanism independent of RNA-based cleavage (*Sampson et al., 2013*). Together with evidence that some Cas9 homologs can target single-stranded DNA substrates under some conditions (*Ma et al., 2015*; *Zhang et al., 2015*), these studies raised the possibility that certain Cas9 enzymes might have intrinsic RNA-guided RNA cleavage activity.

To determine whether evolutionarily divergent Cas9 homologs have a native capacity for programmable RNA targeting, we compared biochemical behavior of enzymes from the three major Cas9 subtypes. This analysis revealed that certain type II-A and II-C Cas9s can bind and cleave single-stranded RNA sequences with no requirement for a PAM or PAMmer. Furthermore, we found that this activity can inhibit gene expression and confer moderate protection against infection by ssRNA phage through a mechanism reminiscent of RNA-guided DNA targeting. These results establish the utility of Cas9 for facile RNA-guided RNA targeting and suggest that this activity may have biological relevance in bacteria.

## Results

### Cas9 catalyzes PAM-independent RNA-guided RNA cleavage

To assess whether divergent Cas9 enzymes can catalyze binding to and cleavage of RNA substrates by a mechanism distinct from that of double-stranded DNA cleavage, we tested homologs from the three major subtypes of Cas9 proteins for their ability to cleave single-stranded RNA in vitro (*Figure 1A,B*; *Figure 1—figure supplement 1A–C*). When programmed with a cognate sgRNA, *S. aureus* Cas9 (SauCas9) and *C. jejuni* Cas9 (CjeCas9) direct cleavage of RNA in the absence of a PAMmer (*Figure 1*; *Figure 1—figure supplement 1*). No RNA cleavage was detected using Spy-Cas9, which requires a PAMmer for efficient RNA cleavage in vitro (*O'Connell et al., 2014*), or using *F. novicida* Cas9 (FnoCas9). While the cleavage efficiencies for both SauCas9 and CjeCas9 are indistinguishable (*Figure 1—figure supplement 1D*), we focused on the activity of SauCas9 due to the abundance of mechanistic and structural data for this enzyme (*Nishimasu et al., 2015*; *Friedland et al., 2015*; *Ran et al., 2015*; *Kleinstiver et al., 2015*).

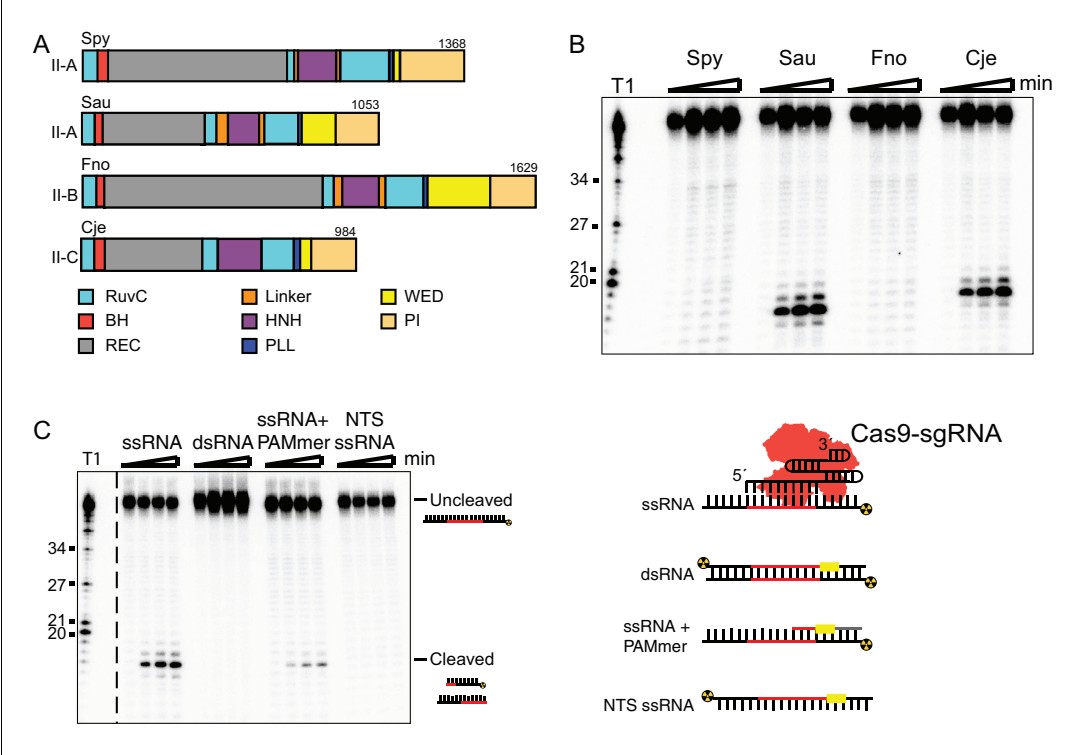

**Figure 1.** SauCas9 cleaves single-stranded RNA without a PAMmer. (**A**) Schematic of Cas9 proteins tested for sgRNA-mediated RNA cleavage. RuvC, RuvC nuclease domain; BH, bridge-helix; REC, recognition domain; HNH, HNH nuclease domain; PLL, phosphate-lock loop; WED, wedge domain; PI, PAM-interacting domain. Adapted from (*Nishimasu et al., 2014*; *2015*; *Hirano et al., 2016*; *Yamada et al., 2017*). (**B**) Representative in vitro cleavage of ssRNA by Cas9-sgRNA RNP complexes of homologs in (**A**). Radiolabeled pUC target RNA was incubated with Cas9 RNP at 37°C and time points were taken at 0, 10, 30, and 60 min. Full time course is presented in *Figure 1—figure supplement 1B*. T1 indicates size markers generated by RNase T1 digestion of ssRNA target. Size in nucleotides is indicated on the left. (**C**) (Right) In vitro cleavage assay of various RNA substrates (Left). Full time course is presented in *Figure 1—figure supplement 3A*.

DOI: https://doi.org/10.7554/eLife.32724.003

The following source data and figure supplements are available for figure 1:

**Figure supplement 1.** RNA is cleaved by SauCas9 and CjeCas9.
DOI: https://doi.org/10.7554/eLife.32724.004

**Figure supplement 1—source data 1.** ssRNA cleavage time course for Cas9 homologs.
DOI: https://doi.org/10.7554/eLife.32724.005

**Figure supplement 2.** ssRNA cleavage is similar to canonical dsDNA cleavage by Cas9.
DOI: https://doi.org/10.7554/eLife.32724.006

**Figure supplement 2—source data 1.** Quantification of multi-turnover cleavage.
DOI: https://doi.org/10.7554/eLife.32724.007

**Figure supplement 3.** SauCas9 cleavage of different nucleic acid substrates.
DOI: https://doi.org/10.7554/eLife.32724.008

**Figure supplement 3—source data 1.** Quantification of SauCas9 cleavage of nucleic acid substrates.
DOI: https://doi.org/10.7554/eLife.32724.009

**Figure supplement 4.** SauCas9 prefers a complementary region of 23nt for binding and cleavage.
DOI: https://doi.org/10.7554/eLife.32724.010

**Figure supplement 4—source data 1.** Cleavage and binding data for different length guides.
DOI: https://doi.org/10.7554/eLife.32724.011

RNA cleavage activity and products were similar to those of canonical Cas9-mediated DNA cleavage activity in vitro. RNA targeting by SauCas9 requires the presence of a guide RNA and a catalytically-active protein, as both apo protein lacking the guide RNA and a catalytically inactive mutant (D10A and N580A) do not cleave RNA (*Figure 1—figure supplement 2A*). Furthermore, addition of EDTA to chelate divalent metal ions abolished RNA cleavage, verifying that divalent metal ions are

necessary for catalysis. As with DNA substrates (*Sternberg et al., 2014*), incubation of SauCas9 with an excess of RNA target demonstrated that cleavage is single-turnover (*Figure 1—figure supplement 2B,C*). Hydrolysis mapping of the cleavage product revealed that the predominant RNA cleavage site is shifted by one nucleotide compared to the site of DNA cleavage (*Garneau et al., 2010*; *Jinek et al., 2012*; *Gasiunas et al., 2012*) (*Figure 1—figure supplement 2D,E*). The shift is consistent with that observed for PAM-dependent SpyCas9 RNA-cleavage (*O'Connell et al., 2014*) and is likely due to the more compact geometry of an RNA-RNA helix relative to an RNA-DNA hybrid helix (*Wang et al., 1982*).

SauCas9 targets ssRNA in the absence of a PAMmer, a contrast to SpyCas9 targeting of ssRNA (*O'Connell et al., 2014*). Testing SauCas9 in vitro ssRNA cleavage in the presence of a PAMmer (30x molar excess over ssRNA target) revealed that turn-over was two-fold slower than the reaction with only target ssRNA (*Figure 1C*, *Figure 1—figure supplement 3C*). SauCas9 ssRNA cleavage conducted in the presence of a non-complementary, control DNA oligo did not yield a similar reduction in cleavage rate (*Figure 1—figure supplement 3C*), indicating that the complementary PAMmer impairs RNA cleavage activity. Consistent with cleavage being guide-dependent, single-stranded RNA that is not complementary to the sgRNA is not cleaved (*Figure 1* and *Figure 1—figure supplement 3*). Double-stranded RNA (dsRNA) is also not a substrate for SauCas9.

Given that Cas9 proteins are active with different length guide RNA segments (~20–24 nt) (*Chylinski et al., 2013*; *Ran et al., 2015*; *Friedland et al., 2015*; *Kim et al., 2017*), we tested whether longer guide segments might enhance ssRNA targeting activity. Increasing the length of the targeting region of the guide up to 23 nt results in tighter binding and more efficient cleavage (*Figure 1—figure supplement 4*), mirroring the preference for longer guides for DNA cleavage (*Ran et al., 2015*; *Friedland et al., 2015*). Extending the guide strand complementarity to the target beyond 23 nt did not increase RNA target binding or cleavage efficiency, indicating that 23 nt is the optimal length for in vitro binding and targeting applications. The apparent dissociation constant ($K_{d,app}$) of the SauCas9-sgRNA complex (23 nt targeting region) for the ssRNA target is 1.8 ± 0.09 nM (*Figure 1—figure supplement 4D*), which is ~5 x weaker than the 0.34 ± 0.03 nM binding affinity measured for a dsDNA substrate of the same sequence.

## Cleavage efficiency is impaired by duplex regions in target RNA

We noted that SauCas9-catalyzed ssRNA cleavage is limited to ~30% fraction cleaved (*Figure 1—figure supplement 3*), compared to >80% fraction cleaved for ssDNA and dsDNA targets. Greater thermodynamic stability of RNA secondary structures, relative to those in ssDNA (*Bercy and Bockelmann, 2015*), might occlude SauCas9-sgRNA binding to an ssRNA target sequence, a possibility that we tested using a panel of partially duplexed RNA substrates (*Figure 2*). Previously, introduction of a short segment of mismatched base pairs to mimic partially unwound dsDNA substrates was shown to enhance the ability of type II-C Cas9s (including CjeCas9) to unwind and cleave dsDNA (*Ma et al., 2015*). Here, we found that duplex-RNA substrates containing a 2- or 6-base pair mismatched segment located near the 5' or 3' end of the 23 nt guide RNA region of the sgRNA could not be cleaved (*Figure 2A–C*, substrates 5, 6, 10, and 11). However, when the unpaired region was increased to 12-base pairs, SauCas9 was able to cleave the target strand. There was a slight cleavage preference for RNA substrates in which the 12-base pair mismatched segment is located near the 5' end of the guide sequence of the sgRNA (*Figure 2A–C*, substrates 7 and 12).

Interestingly, the 23-base pair mismatched segment RNA substrates ('Bubble' substrates 8 and 9) are targeted more efficiently than their ssRNA counterparts (substrates 1 and 2) (*Figure 2C*). We measured the binding affinity of all substrates and found that both the 23-base pair mismatched segment RNA and ssRNA substrates are bound with similar affinity (*Figure 2D*). Furthermore, the apparent difference in cleavage efficiency was not due to the presence of a double-stranded PAM sequence, as mutating the PAM region does not impair cleavage (*Figure 2C*, compare substrates 8 and 9). We hypothesize that RNA containing a mismatched segment presents a more accessible substrate to the Cas9-sgRNA complex due to stable annealing between the ends of the non-target and target strands, whereas the ssRNA substrate alone has ends that are predicted to stabilize a conformation that is partially structured and therefore inaccessible (*Figure 2—figure supplement 1A*).

An alternative hypothesis to explain the limited cleavage of ssRNA substrates is that SauCas9 enzyme inactivation occurs over the course of the reaction, even with SauCas9 protein-sgRNA (ribonucleoprotein, RNP) present in 10-fold excess relative to the ssRNA substrate. To test this, we

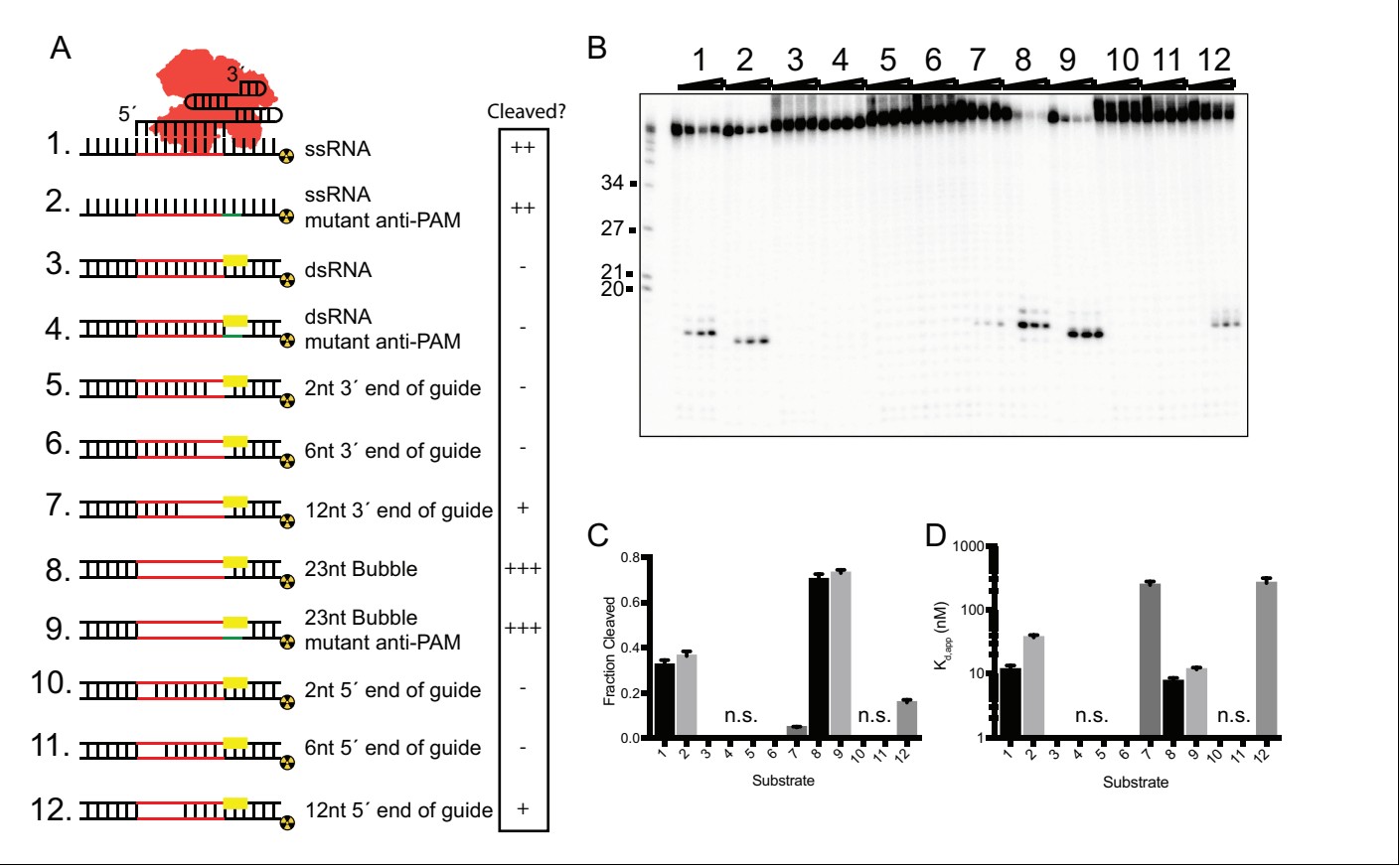

**Figure 2.** In vitro RNA cleavage is impaired by strong secondary structure. (**A**) Schematic representation of structured RNA targets for in vitro cleavage assays. Symbols on right indicate relative level of cleavage activity for each substrate: '-", no cleavage; '+", low cleavage; '++", medium cleavage; '+++" high cleavage. (**B**) Representative cleavage assay of partially-duplexed RNA targets diagrammed in (**A**). T1 indicates size markers generated by RNase T1 digestion of ssRNA target. Size in nucleotides is indicated on the left. (**C and D**) Fraction of target cleaved (**C**) and $K_{d,app}$ (**D**) for substrates diagrammed in (**A**). Fits were determined in Prism using a single-exponential decay and a one-site binding model, respectively. Bars represent mean ± S.D. (n = 3). N.s. denotes no significant cleavage or binding.
DOI: https://doi.org/10.7554/eLife.32724.012

The following source data and figure supplements are available for figure 2:

**Source data 1.** Quantification of cleavage and binding of structured RNA substrates by SauCas9.
DOI: https://doi.org/10.7554/eLife.32724.015

**Figure supplement 1.** RNA cleavage is limited by the RNA target.
DOI: https://doi.org/10.7554/eLife.32724.013

**Figure supplement 1—source data 1.** Quantification of ssRNA after additional protein and target spike-in.
DOI: https://doi.org/10.7554/eLife.32724.014

spiked reactions with fresh SauCas9 protein alone or SauCas9 RNP after reactions reached equilibrium; however, we did not observe an increase in the amount of ssRNA cleavage (*Figure 2—figure supplement 1B,C*). We also tested whether the SauCas9 RNP was able to cleave a second ssRNA substrate that was added to the reaction after it reached completion (*Figure 2—figure supplement 1D,E*). After 1 hr of incubation, the addition of a second target ssRNA complementary to the guide RNA resulted in a burst of cleavage activity, whereas a non-complementary ssRNA substrate did not stimulate cleavage. The second target ssRNA is cleaved to a comparable extent to that observed when this second target was the only substrate in the reaction (*Figure 2—figure supplement 1D,E*, compare reactions 1 and 3). These observations suggest that SauCas9 RNP is still competent and available for cleavage at the end of the reaction and that a property intrinsic to the ssRNA substrate is the limiting factor. We propose that the observed difference in cleavage extents for various RNA

substrates reflects the fraction of molecules that are structurally accessible for cleavage by the Sau-Cas9 RNP.

## SauCas9 confers in vivo protection against RNA phage

Based on the biochemical ability of SauCas9 RNP to bind and cleave ssRNA substrates, we wondered whether this activity might provide protection against RNA phage infection in bacteria. To test this, we generated a plasmid library encoding sgRNAs containing guide sequences complementary to the genome of MS2, a single-stranded RNA phage that can infect *E. coli*. A subset of these sgRNAs contained scrambled guide sequences that should not target MS2, providing negative controls. Another sgRNA subset included single-nucleotide mismatches introduced at each position of a target sequence to test for mismatch sensitivity in ssRNA recognition. This plasmid library, comprising 18,114 sgRNAs, was co-transformed into *E. coli* along with a vector encoding a catalytically active version of SauCas9 and the population of transformants was subjected to infection by bacteriophage MS2 (*Figure 3A*). The experiment was performed in biological triplicate and included an untreated control population and two experimental conditions (multiplicities of infection (MOIs) of 10 and 100). After selection, plasmids were recovered from surviving colonies and sequenced (*Figure 3A*).

We identified between 131 and 166 sgRNAs that were significantly enriched (false discovery rate (FDR)-adjusted p-value<0.05) in the two different MS2 infection conditions (*Figure 3B*). The majority of these sgRNAs were perfectly complementary to the MS2 genome, and only three and five control sgRNAs (out of 708 total control sgRNAs) for the MOI-10 and −100 conditions, respectively, were enriched (*Figure 3B*). The lengths of enriched guide sequences were skewed toward shorter targeting lengths (*Figure 3—figure supplement 1A*, left); however, this likely reflects bias in the cloned input library since the ratio between the enriched guide sequences and those of the library without phage selection are similar (*Figure 3—figure supplement 1A*, right). When comparing the degree of enrichment between the different guide lengths, the 23-nt guide segment sgRNAs were preferentially enriched over those of shorter length (*Figure 3C*), consistent with the in vitro observation that longer guides are more efficient for directing ssRNA cleavage (*Figure 1—figure supplement 4C*). To assess whether there was any sequence bias within the enriched guides, we aligned guide sequences of all lengths at their 3' end. These alignments showed no specific sequence bias in the enriched guides relative to those in the unselected library (*Figure 3—figure supplement 1B*). This is consistent with the crystal structure of an SauCas9-sgRNA-DNA-bound complex which revealed the absence of base-specific contacts of Cas9 to the target strand (*Nishimasu et al., 2015*).

Strikingly, mapping enriched guide sequences onto the MS2 genome showed that enriched sgRNAs were clustered at specific regions, which were consistent across both experimental conditions (*Figure 3D*; *Figure 3—figure supplement 1C,D*). Together with our biochemical data suggesting that SauCas9 cannot bind or cleave structured RNAs (*Figure 3*), we interpret these targeting 'hotspots' to be regions of low structural complexity. It is important to note that sgRNAs containing different guide segment lengths overlap at these regions, possibly indicating that increases in targeting efficiency due to guide length are secondary to target accessibility to the Cas9 RNP. We mapped the enriched guide sequences onto the published secondary structure of the MS2 genome determined through cryoelectron microscopy (*Dai et al., 2017*) (*Figure 3—figure supplement 2*). Guides targeted not only single-stranded, accessible regions but also those that form apparently stable secondary structures. The structure of the MS2 genome was determined on the intact phage particle, however, and may not represent the RNA structure(s) relevant to the infection stage during which SauCas9-mediated protection is crucial.

Highly enriched sgRNAs from the screen were confirmed for their ability to confer protection against MS2 phage infection through a soft-agar plaque assay. Reconstitution of SauCas9 with a targeting guide confers approximately a ten-fold protection against the RNA phage (*Figure 3E,F*). No protection was observed in the absence of an sgRNA or SauCas9 protein. Scrambling the sequence of the guide also abrogates protection, confirming that sequence complementary is necessary for phage elimination. Guide segments of all lengths tested (20–23 nts) conferred protection to a similar level (*Figure 3—figure supplement 3A,B*), consistent with the result from the MS2 screen that guide segments of all lengths were enriched in 'hotspot' regions (*Figure 3D*; *Figure 3—figure supplement 1C*). Two 'control' guides were enriched in both the MOI-10 and −100 treatments. Interestingly, both guides conferred protection but their scrambled counterparts did not (*Figure 3—figure*

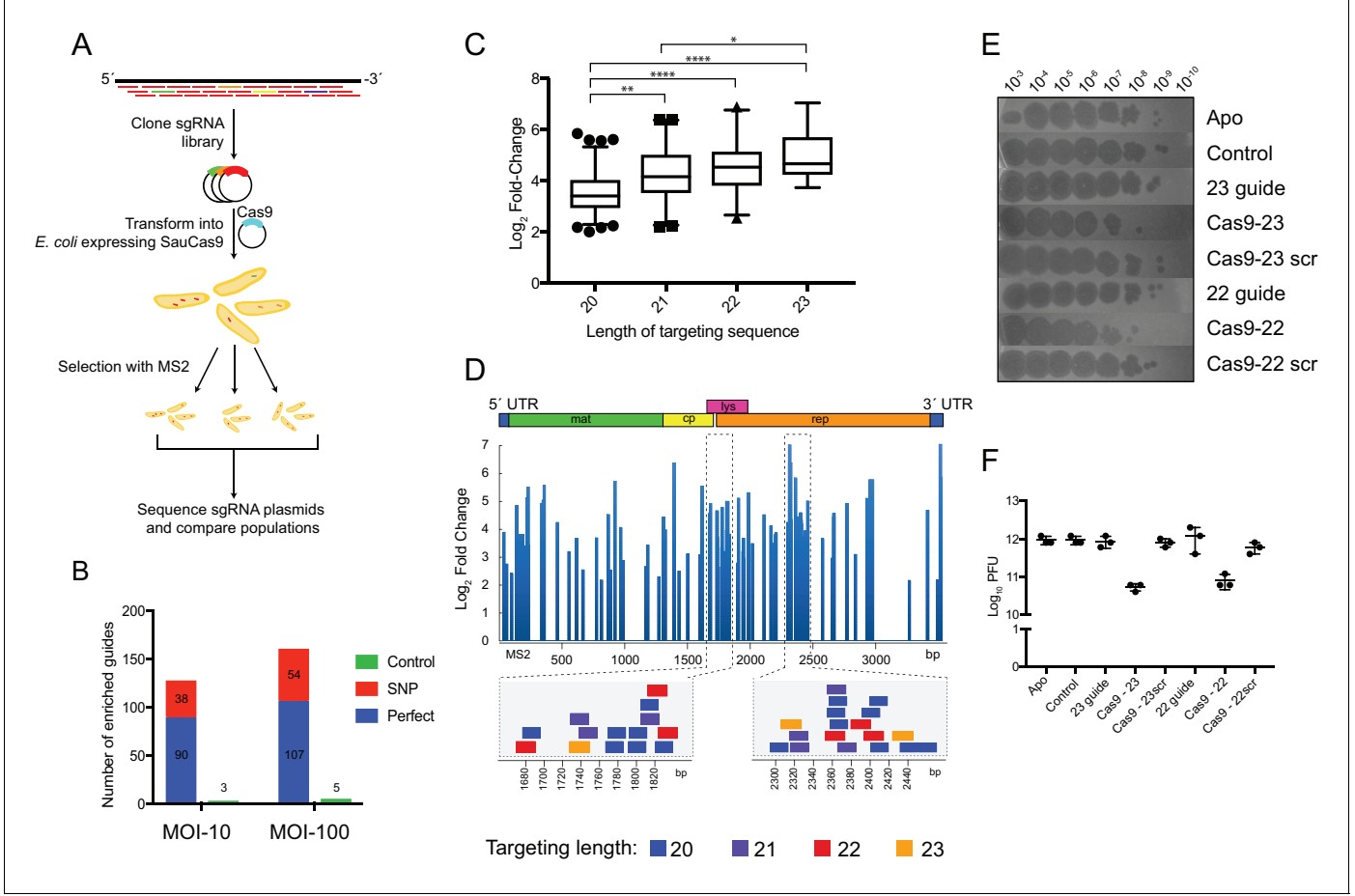

**Figure 3.** SauCas9 confers in vivo protection against an RNA phage. (**A**) Overview of MS2 targeting screen. Guides tiled against the library were cloned into sgRNA expression plasmids and co-transformed into *E. coli* with a plasmid containing wild-type SauCas9 under inducible control. Plasmids from surviving colonies after MS2 selection were recovered and sequenced. For more detail, see Materials and methods. (**B**) Number of guides with significant positive enrichment from three biological experiments. SNP, guides with single-nucleotide mismatch. (**C**) Box and whiskers plot of average $\log_2$ fold-change of perfect MS2 guides by length. Whiskers represent 5% and 95% values with outliers graphed as points. *$p<0.05$, **$p<0.01$, ****$p<0.0001$, by one-way ANOVA. (**D**) (Upper) $\log_2$ fold-change of guides with an FDR-corrected p-value$<0.05$ mapped to the MS2 genome for MOI-100 treatment. Schematic of MS2 genome is provided above. (Lower) Individual guides mapped to highlighted regions of MS2 genome. Other graphs for MOI-10 and −100 treatments are presented in *Figure 3—figure supplement 1*. (**E**) Representative plaque assay of SauCas9 in vivo protection. *E. coli* containing constructs on the right are spotted with various phage dilutions as indicated. Scr signifies that the targeting portion of the guide has been scrambled to serve as a non-targeting control. (**F**) Relative plaque forming units (PFU) (mean ± S.D., n = 3) from results in (**E**). More guides and controls are presented in *Figure 3—figure supplement 3*.

DOI: https://doi.org/10.7554/eLife.32724.016

The following source data and figure supplements are available for figure 3:

**Source data 1.** List of guides, normalized read counts, enriched guides, fold change distribution and plaque enumeration.

DOI: https://doi.org/10.7554/eLife.32724.025

**Figure supplement 1.** Enriched guides do not display sequence bias and cluster to regions on the MS2 genome.

DOI: https://doi.org/10.7554/eLife.32724.017

**Figure supplement 1—source data 1.** Enriched guide length distribution, sequences, and targeting location on MS2 genome.

DOI: https://doi.org/10.7554/eLife.32724.018

**Figure supplement 2.** Enriched MS2 targeting guides mapped to MS2 genome structure.

DOI: https://doi.org/10.7554/eLife.32724.019

**Figure supplement 2—source data 1.** Location of enriched guides from MOI-100 condition mapped to MS2 genome.

DOI: https://doi.org/10.7554/eLife.32724.020

**Figure supplement 3.** Confirmation that enriched guides from the MS2 screen confer protection against MS2 infection.

DOI: https://doi.org/10.7554/eLife.32724.021

**Figure supplement 3—source data 1.** Plaque enumeration for SauCas9-mediated MS2 protection.

*Figure 3 continued*

DOI: https://doi.org/10.7554/eLife.32724.022

**Figure supplement 4.** Effect of single-nucleotide mismatches on ssRNA targeting.

DOI: https://doi.org/10.7554/eLife.32724.023

**Figure supplement 4—source data 1.** Heatmaps of single-nucleotide mismatches from MS2 screen and in vitro mismatch cleavage.

DOI: https://doi.org/10.7554/eLife.32724.024

supplement 3C,D). Whereas a possible off-target binding site was found for one guide (#14238) within the MS2 genome (*Figure 3—figure supplement 3E*), it remains unclear how guide #14210 confers protection. Possibly this sgRNA acts by targeting an *E. coli* host factor that is necessary for infection.

Screening against the MS2 genome was also used to test the effect of single-nucleotide mismatches on SauCas9's targeting ability. We computed an average fold change (between phage treated and untreated samples) for all sgRNAs that contained a mismatch at the same position, and obtained average values for mismatches at each position across the guide. We observed a pronounced gradient of increasing guide stringency with length. On average, short guides were less sensitive to mismatches, while mismatches in longer sgRNAs led to decreased recovery compared to control samples (*Figure 3—figure supplement 4A,B*). Previous work and models suggest that shorter guide segments should be more sensitive to mismatches and lead to higher fidelity Cas9 targeting (*Fu et al., 2014*; *Bisaria et al., 2017*). Further study is needed to thoroughly examine this unexpected pattern of RNA-targeting stringency, as one shortcoming of this experiment is that mismatched guides were not designed, *a priori*, to recognize accessible parts of the MS2 genome. Nevertheless, despite potential noise introduced in this analysis due to guide segments that target inaccessible MS2 regions, we observe an interesting correlation between mismatches in the MS2 screen and in vitro biochemical cleavage assays for the sgRNA with a 23 nt guide segment sequence (*Figure 3—figure supplement 4C,D*). The first few nucleotides in the 'seed' region (guide 3′ end proximal) are sensitive to mismatches, while a central region of sensitivity is also observed, similar to previously demonstrated regions of sensitivity for SpyCas9 DNA cleavage (*Cong et al., 2013*; *Jiang et al., 2013*; *Fu et al., 2016*; *Gorski et al., 2017*).

## SauCas9 represses gene expression in *E. coli*

An efficient RNA-targeting Cas9 could serve as an important tool in regulating gene expression in vivo. To test the ability of SauCas9 to mediate repression of host gene expression, we targeted dSauCas9 and dSpyCas9 RNPs to a GFP reporter sequence encoded in the *E. coli* chromosome (*Qi et al., 2013*). Catalytically inactive versions of Cas9 were used to prevent cleavage of the bacterial chromosome when targeting a site adjacent to a PAM. As expression of Cas9 and sgRNA exerts metabolic stress on *E. coli*, GFP fluorescence values were normalized by the $OD_{600}$ value to account for differences in cell growth between cultures (*Oakes et al., 2016*). When using sgRNAs designed to recognize a sequence in the GFP gene adjacent to the appropriate PAM for SauCas9 (NNGRRT) or SpyCas9 (NGG), GFP expression is significantly reduced (*Figure 4A*) consistent with CRISPR-interference (CRISPRi) (*Qi et al., 2013*; *Gilbert et al., 2014*). When sgRNAs were designed to recognize GFP sequences not flanked by a PAM, dSauCas9 but not dSpyCas9 was able to repress GFP expression. The SauCas9-mediated GFP repression was dependent on sgRNAs that target the coding strand; sgRNAs that recognize the non-coding strand did not result in reduced GFP expression (*Figure 4—figure supplement 1A*). The length of the targeting sequence in vivo corroborates in vitro data, with longer guides working more efficiently (*Figure 4B*).

Different guide sequences display variable efficiencies of targeting. We tiled sgRNAs across the GFP mRNA sequence to test the robustness of dSauCas9 to repress GFP expression (*Figure 4C*). As no sites are adjacent to PAM sequences, all repression presumably occurs on the mRNA level. The efficiency of dSauCas9-mediated GFP repression varied according to the target sequence, with some dSauCas9 RNPs reducing GFP signal to 15–30% of that observed in the presence of the sgRNA alone (*Figure 4C*, GFP2 and 6) and others showing no ability to repress GFP expression (GFP7 and 9). Electrophoretic mobility shift assays support the conclusion that repression is not occurring at the dsDNA level by promiscuous PAM binding (*Figure 4—figure supplement 1B*).

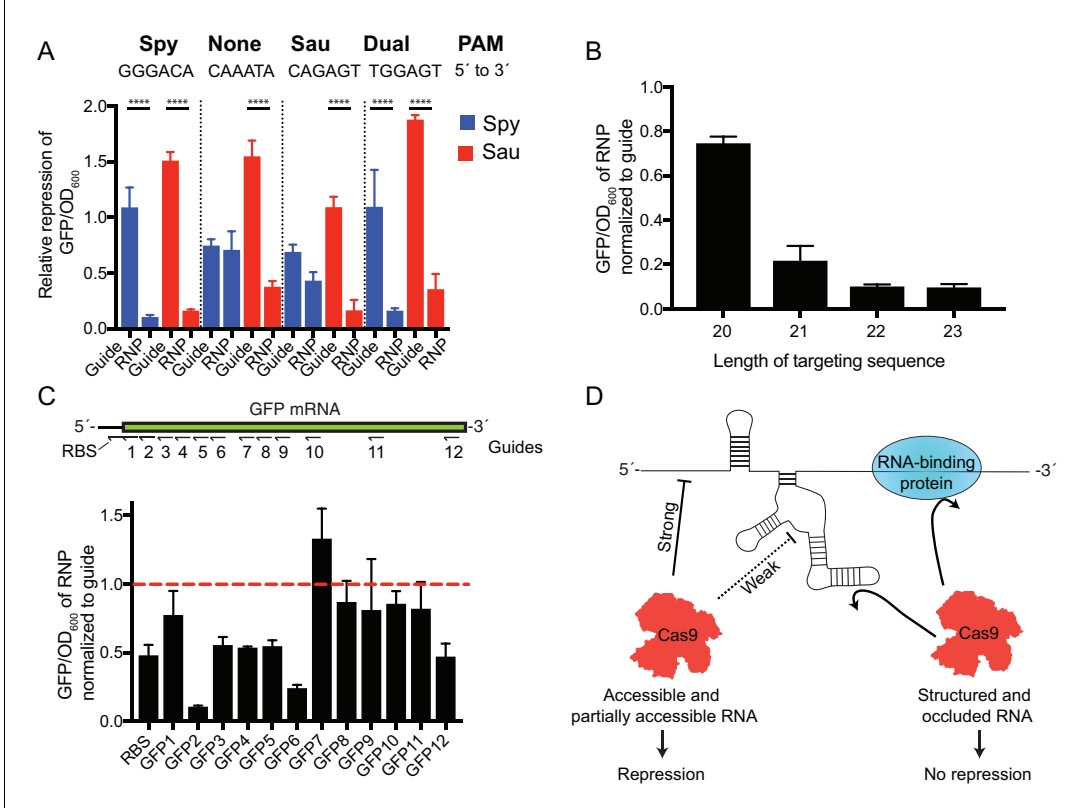

**Figure 4.** SauCas9 repression of a GFP reporter in vivo. (**A**) Comparison of dSpy and dSauCas9 to repress GFP expression on the DNA and RNA level. GFP signal is normalized to $OD_{600}$ to control for difference in cell density between samples. $GFP/OD_{600}$ ratios for guide alone and RNP are normalized to values for a non-targeting guide vector and an Apo protein control, respectively. Target sites were chosen to be adjacent to PAM sites for Spy, Sau, both, or neither as indicated. Note: the slight GFP repression observed with dSpyCas9 using the target sequence adjacent to the Sau PAM (CAGAGT) likely results from the ability of SpyCas9 to use an NAG PAM, albeit with reduced efficiency (**Hsu et al., 2013**). ****$p<0.0001$ by one-way ANOVA. (**B**) Relative expression of GFP using guides with different length targeting sequences. Target site here is the GFP2 sequence chosen for its robust targeting activity. (**C**) (Upper) Diagram of targeting sequences across the GFP mRNA and ribosome-binding site (RBS). (Lower) Relative expression of GFP of SauCas9 RNP normalized to sgRNA alone for targeting sequences across the GFP reporter. Dashed red line indicates that the sgRNA alone is as efficient as the RNP for GFP repression. (**A–C**) Bars represent mean ± S.D. (n = 3). (**D**) Model for observed SauCas9 ssRNA targeting activity. We propose that accessible RNA is cleaved or repressed efficiently while structured and protein-bound RNA is not targeted by SauCas9.

DOI: https://doi.org/10.7554/eLife.32724.026

The following source data and figure supplements are available for figure 4:

**Source data 1.** Raw data for PAM dependency, length efficiency, and GFP mRNA tiling for GFP repression assays.
DOI: https://doi.org/10.7554/eLife.32724.029

**Figure supplement 1.** Repression of GFP mRNA.
DOI: https://doi.org/10.7554/eLife.32724.027

**Figure supplement 1—source data 1.** Raw data for GFP repression assays.
DOI: https://doi.org/10.7554/eLife.32724.028

Repression is largely equivalent between catalytically active and inactive forms of SauCas9 (**Figure 4—figure supplement 1C**), suggesting that binding of the Cas9-sgRNA complex to the mRNA is sufficient for repression and consistent with in vitro data showing that the enzyme does not catalyze multiple-turnover RNA cleavage. While we speculate that the Cas9-RNP blocks the ribosome directly (either at initiation or during elongation), our data do not rule out the possibility that Cas9 is otherwise destabilizing the mRNA transcript through an unknown mechanism.

Together our biochemical and in vivo data support a model in which SauCas9 can readily bind and cleave bacteriophage RNA and mRNA sequences that are exposed and unstructured (**Figure 4D**). Regions that form strong structures are inaccessible to SauCas9 RNP binding, thereby

preventing cleavage or repression activity. As Cas9 cleavage activity is limited by target accessibility, we expect that RNA occluded by RNA-binding proteins would also be recalcitrant to cleavage.

## Discussion

Investigation of CRISPR-Cas9 has focused on its function as a double-stranded DNA endonuclease, while the ability of diverse homologs to cleave natural RNA substrates has remained unexplored. Here, we present evidence that type II-A and type II-C Cas9 enzymes can catalyze programmable and PAM-independent single-stranded RNA cleavage. Focusing on SauCas9, we show that this enzyme can be employed both biochemically and in cells to cleave RNA and regulate genes on both the transcriptional and translational level in parallel by accounting for target site PAM proximity. Importantly, SauCas9 ssRNA scission requires only an sgRNA and does not need a PAMmer, thereby simplifying applications (*Nelles et al., 2015*) and facilitating delivery to cells as a pre-assembled RNP (*Zuris et al., 2015*; *Mout et al., 2017*)

The RNA-targeting capability of SauCas9 and related Cas9 enzymes offers the advantage of repressing viruses whose lifecycles do not involve a DNA genome or intermediate, thereby rendering them inaccessible to Cas9-mediated DNA cleavage. We demonstrated that SauCas9 could be programmed to confer protection to *E. coli* against MS2, an RNA bacteriophage with no DNA intermediate. Whether RNA-based viral repression by Cas9 occurs in natural systems is not known, but seems possible based on our results. DNA cleavage by SauCas9 remains more rapid than RNA cleavage, indicating that DNA-targeting is probably the biologically preferred method for phage and plasmid interference. However, Cas9 activity on RNA is PAM-independent and may mitigate the effects of PAM-escape mutants that would evade DNA-level interference (*Deveau et al., 2008*), thus acting as an additional line of defense.

Intriguingly, 'hotspots' of preferential targeting emerged when tiling guides across the genome, but these sites were devoid of sequence bias. In conjunction with in vitro cleavage data of partially structured RNAs, we suggest that SauCas9 cleavage efficiency is inversely related to structural complexity of the RNA target. As an alternative to the current approach of screening multiple sgRNAs for activity, experimental knowledge about RNA structure, such as SHAPE-seq data (*Loughrey et al., 2014*), would simplify target identification for viral targeting and repression experiments. Nevertheless, future work will concentrate on understanding the structural constraints on RNA targeting and methods to improve Cas9 access to duplex RNA regions.

SauCas9 holds promise for a range of RNA targeting applications. We showed that SauCas9 could repress gene expression in *E. coli*. Repression of the reporter occurs in the absence of the PAM and is specific for targeting of the coding strand. Recently, the Type VI CRISPR-Cas system effector, Cas13, has been proposed and demonstrated to target RNA (*Shmakov et al., 2015*; *Abudayyeh et al., 2016*; *East-Seletsky et al., 2016*). 'Activated' Cas13 exhibits robust *trans* cleavage of RNAs(*Abudayyeh et al., 2016*; *East-Seletsky et al., 2016*; *Smargon et al., 2017*). While RNA-cleavage by SauCas9 is single-turnover and kinetically less robust than that of Cas13, Cas9 does not cleave RNAs indiscriminately and lends itself to targeting of specific transcripts. A programmable Cas9 capable of repressing genes on the RNA level has potential advantages over CRISPRi DNA-based techniques (*Qi et al., 2013*; *Gilbert et al., 2014*). For example, isoform-specific targeting of different transcripts originating from the same transcription start site or resulting from alternative splicing events might be possible. More broadly, due to its intrinsic ssRNA-binding activity, SauCas9 may have utility as a platform for directing other effector proteins to specific RNA molecules, such as proteins or domains that up-regulate translation or RNA base-modifying enzymes for site-specific epigenetic modification of RNAs.

## Materials and methods

**Key resources table**

| Reagent type (species) or resource | Designation | Source or reference | Identifiers | Additional information |
|---|---|---|---|---|
| Strain, strain background (*E. coli*) | BL21(DE3) | Thermo Fisher | | |

*Continued on next page*

*Continued*

| Reagent type (species) or resource | Designation | Source or reference | Identifiers | Additional information |
|---|---|---|---|---|
| Strain, strain background (*E. coli*) | XL1-Blue | QB3-MacroLab | | |
| Strain, strain background (*E. coli*) | strain with GFP reporter | PMID: 27136077 | | |
| Recombinant DNA reagent | SauCas9 expression vector | this paper | | SauCas9 sequence in vector backbone from PMID: 27136077 |
| Recombinant DNA reagent | His-MBP vector (plasmid - #29706) | addgene | | |
| Recombinant DNA reagent | SauCas9 guide expression vector | this paper | | SauCas9 guide scaffold in vector backbone from PMID: 27136077 |
| Software, algorithm | ImageQuantTL | GE Healthcare | RRID:SCR_014246 | |
| Software, algorithm | Prism7 | GE Healthcare | RRID:SCR_015807 | |

## Phylogenetic tree construction and RNA folding

Cas9 homolog sequences were obtained from Chylinski and colleagues (*Chylinski et al., 2014*). A structure-guided alignment was produced using PROMALS3D (*Pei et al., 2008*) and a maximum-likelihood tree was inferred using PHYML3.0 (*Guindon et al., 2010*). The structure of the pUC ssRNA target was predicted using Mfold (*Zuker, 2003*).

## Protein purification

All proteins were expressed as His-MBP fusions (Addgene vector #29706) in *E. coli* strain BL21(DE3). Cells were grown to an $OD_{600}$ of 0.6–0.8, induced with 0.4M IPTG, and then incubated overnight at 16°C with shaking. Proteins were purified using Superflow Ni-NTA affinity resin (Qiagen, Valencia, CA), followed by a HiTrap HP Heparin column (GE Healthcare, Pittsburgh, PA) and gel filtration on a Superdex S200 (GE Healthcare, Pittsburgh, PA), as previously described (*Jinek et al., 2012*). Cas9 protein sequences can be found in *Supplementary file 1*.

## Oligonucleotide purification and radiolabeling

DNA oligonucleotides were synthesized by IDT (Coralville, IA). Target RNAs and sgRNAs were transcribed in vitro as previously described (*Sternberg et al., 2012*). DNA targets and in vitro transcribed RNAs were gel purified by 7M urea denaturing PAGE. Target RNAs and DNAs were 5′ end-labeled with [γ-P32-ATP] by treatment with PNK (NEB, Ipswich, MA). T1 sequencing and hydrolysis ladders were prepared according to manufacturer's directions (Ambion, Grand Island, NY). A list of all sgRNAs and targets can be found in *Supplementary file 1*.

## In vitro cleavage assays

Cas9 was reconstituted with equimolar sgRNA in 1x cleavage buffer (20 mM Tris-HCl – pH 7.5, 200 mM KCl, 1 mM TCEP, 5% glycerol, 5 mM MgCl2) for 10 min at 37°C, then immediately placed on ice. Cleavage reactions were conducted with 1 nM target and 10 nM reconstituted Cas9-sgRNA in 1x cleavage buffer unless otherwise noted. Structured RNA substrates were prepared by annealing two separate in vitro transcribed RNAs. The target strand was annealed with 10-fold excess of the non-target strand to ensure that all target is complexed prior to the cleavage reaction. Reactions were incubated at 37°C for the indicated time and quenched in Heparin-EDTA buffer (10 µg/ml heparin, 25 mM EDTA) at 25°C for 5 min. Reactions were diluted with 2x Formamide loading buffer and incubated at 95°C for 5 min prior to separation on a 15% denaturing 7M urea PAGE gel. Gels were dried overnight and exposed to a phosphor imaging screen (Amersham/GE Healthcare, Pittsburgh, PA). Results were visualized on a Typhoon (GE Healthcare, Pittsburgh, PA) and quantified in ImageQuantTL (v8.1, GE Healthcare, Pittsburgh, PA). The cleaved fraction of total signal was calculated independently for three separate experiments and were fit with a one-phase exponential decay model in Prism7 (GraphPad Software, La Jolla, CA).

## Filter binding and electrophoretic mobility shift assays

Binding reactions consisted of 750 nM catalytically inactive SauCas9 reconstituted with sgRNA to the final concentrations indicated. Radiolabeled target RNA was added to a final concentration of 1

nM and the reactions were incubated at 37°C for one hour. Bound probe was separated from unbound using a three-filter system on a vacuum manifold (*Rio, 2012*). Membranes were allowed to dry prior to phosphor imaging and quantification. EMSAs were performed in the presence of 300 nM dSauCas9 and 1 nM radiolabeled target strand DNA pre-annealed in the presence of 10x non-target strand. Complexes were incubated at 37°C for 1 hr prior to separation on 6% non-denaturing PAGE. Gels were dried prior to phosphor imaging. Three independent experiments were performed and the fraction of bound out of total signal was calculated in ImageQuantTL. Binding isotherms were determined in Prism7 using a one-site binding model.

## MS2 screen and plaque assay

All guides of length 20–23 nt antisense to the MS2 bacteriophage genome were synthesized (CustomArray Inc., Bothell, WA) and cloned into a guide expression vector (*Oakes et al., 2016*) modified with the SauCas9 sgRNA scaffold. XL1-Blue *E. coli* cells with a vector containing a tetracycline-inducible wtSauCas9 construct were made electrocompetent and transformed with the MS2-guide plasmid library in triplicate. Approximately $1 \times 10^6$ transformants were grown for 30 min at 37°C with shaking prior to addition of antibiotics and 10 nM anhydrotetracycline (aTc) (Sigma, St. Louis, MO) for protein induction. After an additional 30 min of growth, cultures were split into three equal pools and treated with none, $3.3 \times 10^6$, or $3.3 \times 10^7$ MS2 bacteriophage. After 3 hr of infection, cells were plated on LB-agar supplemented with antibiotics and incubated at 37°C for 16 hr. Plates were scraped with LB and plasmids were isolated using a MidiPrep kit (Qiagen, Valencia, CA), according to the manufacturer's protocol. High-throughput sequencing libraries were prepared by PCR amplification of the variable region of the guide plasmid. Dual unique-molecular identifiers (UMIs), included to separate true single-nucleotide mismatches, as well as duplicates, from PCR artifacts (*Kou et al., 2016*), were incorporated during a single round of PCR. Excess UMIs were removed by ExoI digestion (Thermo Scientific, Waltham, MA) prior to library amplification and barcoding. Individual guides (*Supplementary file 1*) were cloned using oligonucleotides synthesized by IDT and co-transformed into XL1-Blue *E. coli* cells with the SauCas9 vector. Resistance to MS2 bacteriophage was conducted using a soft-agar overlay method (*Abudayyeh et al., 2016*) and plaque forming units (PFUs) were calculated. To minimize variability in plaquing efficiency, the same phage dilutions were used for all experiments.

## MS2 survival and mismatch analysis

After applying a low-pass filter, reads were trimmed using cutadapt v. 1.14 (*Martin, 2011*) and paired-end overlapping reads were merged using pandaseq (*Masella et al., 2012*) for error correction. Reads were mapped to the MS2 genome with bowtie2 v2.3.0 (*Langmead and Salzberg, 2012*) using the 'very-sensitive' option and de-duplicated based on the dual-UMI (*Smith et al., 2017*). Feature counts were obtained using HTSeq-count (*Anders et al., 2015*). Differential expression was calculated using standard pipelines implemented in 'edgeR' (*Robinson et al., 2010*; *McCarthy et al., 2012*). Significantly enriched guides were defined as those with an FDR-corrected p-value<0.05. Guides with a positive fold-change compared to the control were mapped to the MS2 genome and visualized using the 'Sushi' package (*Phanstiel et al., 2014*). To examine for nucleotide composition bias, sequences of guides with a significant positive enrichment were aligned at the 3′ end (PAM-proximal) and motifs were analyzed using the WebLogo server (*Crooks et al., 2004*). The distribution of $\log_2$ fold-change values of significantly enriched guides plotted as box and whisker plots in Prism. The secondary structure of the MS2 genome was obtained from (*Dai et al., 2017*) and reads were mapped and visualized in Forna (*Kerpedjiev et al., 2015*). $\log_2$ fold-change values of single-nucleotide mismatch (SNP) guides for each treatment were partitioned by length and averaged at each position. High-throughput sequencing data accompanying this paper are available through the Sequencing Read Archive under the BioProject accession number PRJNA413805.

## *E. coli* in vivo GFP repression

Based on the system outlined previously, SauCas9 was cloned into a tetracycline-inducible vector, while individual guides are under control of a constitutive promoter (*Oakes et al., 2016*). Plasmids were transformed into an *E. coli* strain with a GFP reporter gene integrated into the chromosome (*Qi et al., 2013*). Cultures were grown in M9 medium supplemented with 0.4% w/v glucose to mid-

log phase and diluted to an $OD_{600}$ of 0.05 prior to transfer to a Tecan Microplate reader (Tecan Systems, San Jose, CA). Protein expression was induced with 10 nM aTc. GFP and $OD_{600}$ were measured every ten minutes for at least 18 hr. Curves of GFP expression over time were fit with a logistic growth model in Prism. At 80% of the maximum value, or at least after 16 hr of growth, the GFP signal was normalized by cell density at $OD_{600}$. To account for effects of guide and protein expression, $GFP/OD_{600}$ was normalized to a null guide or null protein culture, respectively. As expression of different guides change GFP expression levels, the ratio between normalized RNP and guide values was taken to allow comparison of RNP-based repression across different guides. All experiments were conducted in triplicate and all graphing and quantitative analyses were conducted in Prism. Guide and target sequences can be found in *Supplementary file 1*.

## Acknowledgements

We thank G Knott and A Tambe for critical reading of the manuscript, members of the Doudna lab for helpful advice, L Gilbert and A Tambe for technical expertise, and N Ma, J Ye, and K Zhou for technical assistance. The Vincent J Coates Genomics Sequencing Laboratory is supported by NIH S10 Instrumentation Grants S10RR029668 and S10RR027303.

## Additional information

### Competing interests

Steven C Strutt, Oscar A Negrete: is listed on a patent application (No. 62598888) related to this work. Jennifer A Doudna: is a co-founder of to Caribou Biosciences, Intellia Therapeutics, and Editas Medicine and a scientific advisor to Caribou, Intellia, eFFECTOR Therapeutics and Driver. JAD is listed on a patent application (No. 62598888) related to this work. The other authors declare that no competing interests exist.

### Funding

| Funder | Grant reference number | Author |
|---|---|---|
| National Science Foundation | MCB-1244557 | Steven C Strutt<br>Jennifer A Doudna |
| Howard Hughes Medical Institute | | Rachel M Torrez<br>Jennifer A Doudna |
| German Academic Exchange Program | | Emine Kaya |
| Laboratory Directed Research and Development | U.S. Department of Energy's National Nuclear Security Administration under contract DE-NA0003525 | Oscar A Negrete |
| Paul Allen Frontiers Science Program | | Jennifer A Doudna |

The funders had no role in study design, data collection and interpretation, or the decision to submit the work for publication.

### Author contributions

Steven C Strutt, Conceptualization, Data curation, Formal analysis, Investigation, Visualization, Methodology, Writing—original draft, Writing—review and editing; Rachel M Torrez, Investigation, Writing—review and editing; Emine Kaya, Resources, Investigation, Writing—review and editing; Oscar A Negrete, Conceptualization, Supervision, Funding acquisition, Writing—review and editing; Jennifer A Doudna, Conceptualization, Supervision, Funding acquisition, Writing—original draft, Writing—review and editing

## Author ORCIDs

Steven C Strutt http://orcid.org/0000-0002-4653-5559
Jennifer A Doudna http://orcid.org/0000-0001-9161-999X

## Decision letter and Author response

Decision letter https://doi.org/10.7554/eLife.32724.035
Author response https://doi.org/10.7554/eLife.32724.036

# Additional files

### Supplementary files

• Supplemental file 1. List of sequences used in this study
DOI: https://doi.org/10.7554/eLife.32724.030

• Transparent reporting form
DOI: https://doi.org/10.7554/eLife.32724.031

### Major datasets

The following dataset was generated:

| Author(s) | Year | Dataset title | Dataset URL | Database, license, and accessibility information |
|---|---|---|---|---|
| Strutt S, Doudna J | 2017 | MS2 CRISPR screen in E. coli | https://www.ncbi.nlm.nih.gov/bioproject/?term=PRJNA413805 | Publicly available at NCBI BioProject (Accession no: PRJNA413805) |

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
