## [Decision Letter]

Thank you for submitting your article "RNA-dependent RNA targeting by CRISPR-Cas9" for consideration by *eLife*. Your article has been favorably evaluated by Gisela Storz (Senior Editor) and three reviewers, one of whom is a member of our Board of Reviewing Editors. The following individual involved in review of your submission has agreed to reveal his identity: Ailong Ke (Reviewer #2).

The reviewers have discussed the reviews with one another and the Reviewing Editor has drafted this decision to help you prepare a revised submission.

Your manuscript presents novel Cas9 RNA targeting activity from diverse Type II-A and II-C CRISPR-Cas systems, and show ssRNA targeting and cleavage in a non-canonical PAM- and PAMmer-independent manner. The reviewers universally agree that this is a valuable addition to the literature, and that some of the findings create opportunities for both expansion of the CRISPR toolbox, and for diversification of applications.

Unless you are keen on performing additional experiments to address specific items listed below, altering the narrative to present the antiviral activity as a proof of concept, and speculating about the mechanism of action driving the translational effect would be acceptable.

Summary:

First, the authors showed that both *S. aureus* (Sau) and *C. jejuni* (Cje) Cas9 can cleave about 30-40% of ssRNA substrates in the absence of a PAMmer (which actually lowered cleavage efficiency) and determined that SauCas9 has roughly 5-fold higher K¬D¬ for ssRNA than dsDNA. Next the authors showed that RNA secondary structure can inhibit cleavage, yet interestingly, introducing mismatches in dsRNA substrates recovered activity and even exceeded the activity towards the ssRNA substrate. SauCas9's ability to cleave RNA also allowed *E. coli* to defend itself versus MS2 infection and was shown to be useful in knocking down GFP expression in a reporter *E. coli* line.

Essential revisions:

The biochemical data provided to establish PAM-independent, RNA-mediated ssRNA targeting is convincing and the narrative articulate. The reliance on guide RNA loading, the presence of active residues (D10 and N580) and divalent metal ions is clear, as well as the impairment by target RNA structural features. The same does not apply to the sections dedicated to defense vs. RNA phages, and to a lesser extent, to the repression of protein translation. In the case of viral RNA defense, while there is admittedly basic proof of concept showing reduction by an order of magnitude, it is unclear how potent and biologically relevant and useful this activity is. The authors may consider testing more guides (perhaps in the vicinity of the most functional guides tested thus far, within "hotspots of low structural complexity"), and may consider testing the combination of multiple guides concurrently (a la CRISPR multiplexed arrays, or with multiple independent guides) to show more potent viral resistance. The authors should readily be able to combine guides 23 and 22 to (hopefully) show enhanced resistance (guides 20 and 21 shown on Figure 3—figure supplement 3 are also candidates). This would enable them to keep their current conclusions about the reduction of viral infection. Another alternative would be to tone down the conclusions provided in the narrative, though this is an appealing and valuable portion of the manuscript.

With regards to repression of protein synthesis in bacteria, the levels observed are likewise somewhat quantitatively underwhelming, but do provide proof of concept preliminary results. Again, the authors should consider performing additional experiments (possibly also based on multiplexing guides to enhance the reduction effect), or alter the conclusiveness of their statements. Perhaps two of some of the most potent guides shown on Figure 4 could be combined. This is also a valuable element of the narrative that would be work substantiating more to support the conclusions presented.

All biochemical experiments were performed on the same pUC RNA target, which ignores the effect that target sequence may have on KD¬, spacer length preference, etc. Additionally, while the pUC RNA is predicted to have secondary structure, there are a number of weak or wobble base pairs. How does RNA sequence change Cas9's kinetic parameters and do RNA targets with stronger secondary structure preclude Cas9 cutting?

The authors demonstrated that GFP expression can be reduced by targeting SauCas9 to GFP transcripts. However, the authors didn't provide sufficient evidence to suggest that repression is on the translational level. Cas9 binding may still act on the transcriptional level by interfering with RNAP progression in a Rho-like fashion, or by destabilizing mRNA transcripts, leading to their degradation. Ribosomal profiling should definitively show that Cas9 diminishes expression by blocking ribosome progression.

The translational repression effect is presumably due to action on ribosomes, possibly prevention of ribosome loading, elongation inhibition, or reduction of tRNA accessibility. mRNA degradation affects translation but is not the translation mechanism per se. While ribosome profiling may require extra efforts and not fit within the scope of the manuscript, the authors may consider it, or possibly testing the approach on a bi-cistronic system: RBS1-A-RBS2-B, and show whether targeting A only affects A. Of course, revising the text to tone down conclusiveness on translation control is also an option.

---

## [Author Response]

Essential revisions:The biochemical data provided to establish PAM-independent, RNA-mediated ssRNA targeting is convincing and the narrative articulate. The reliance on guide RNA loading, the presence of active residues (D10 and N580) and divalent metal ions is clear, as well as the impairment by target RNA structural features. The same does not apply to the sections dedicated to defense vs. RNA phages, and to a lesser extent, to the repression of protein translation. In the case of viral RNA defense, while there is admittedly basic proof of concept showing reduction by an order of magnitude, it is unclear how potent and biologically relevant and useful this activity is. The authors may consider testing more guides (perhaps in the vicinity of the most functional guides tested thus far, within "hotspots of low structural complexity"), and may consider testing the combination of multiple guides concurrently (a la CRISPR multiplexed arrays, or with multiple independent guides) to show more potent viral resistance. The authors should readily be able to combine guides 23 and 22 to (hopefully) show enhanced resistance (guides 20 and 21 shown on Figure 3—figure supplement 3 are also candidates). This would enable them to keep their current conclusions about the reduction of viral infection. Another alternative would be to tone down the conclusions provided in the narrative, though this is an appealing and valuable portion of the manuscript.

The authors thank the reviewers for critical evaluation of this work and thoughtful suggestions of experiments. We agree that the biological significance of SauCas9’s RNA-targeting ability remains unknown and that our data constitute a proof-of-concept. We expanded the Discussion to address the potential significance of this activity in the context of CRISPR interference.

All biochemical experiments were performed on the same pUC RNA target, which ignores the effect that target sequence may have on KD¬, spacer length preference, etc. Additionally, while the pUC RNA is predicted to have secondary structure, there are a number of weak or wobble base pairs. How does RNA sequence change Cas9's kinetic parameters and do RNA targets with stronger secondary structure preclude Cas9 cutting?

After initial screening of several ssRNA targets, we chose the pUC target due to its propensity to being cleaved. Other sequences displayed a lower fraction cleaved at completion of the assay and were predicted to form stronger secondary structures than the pUC target. We believe that RNA structure seems to be a limitation on the ability of SauCas9 to cleave RNA given the in vitro cleavage experiment of partially-duplexed RNAs presented in this manuscript. We imagine that sequences that form stronger secondary structures would lower the rate of cleavage by SauCas9; however, SauCas9 RNA-targeting does not seem to be limited to certain RNA sequences as no preferential motifs were detected in the target sequences enriched in the MS2 in vivo screen.

With regards to repression of protein synthesis in bacteria, the levels observed are likewise somewhat quantitatively underwhelming, but do provide proof of concept preliminary results. Again, the authors should consider performing additional experiments (possibly also based on multiplexing guides to enhance the reduction effect), or alter the conclusiveness of their statements. Perhaps two of some of the most potent guides shown on Figure 4 could be combined. This is also a valuable element of the narrative that would be work substantiating more to support the conclusions presented.The authors demonstrated that GFP expression can be reduced by targeting SauCas9 to GFP transcripts. However, the authors didn't provide sufficient evidence to suggest that repression is on the translational level. Cas9 binding may still act on the transcriptional level by interfering with RNAP progression in a Rho-like fashion, or by destabilizing mRNA transcripts, leading to their degradation. Ribosomal profiling should definitively show that Cas9 diminishes expression by blocking ribosome progression.The translational repression effect is presumably due to action on ribosomes, possibly prevention of ribosome loading, elongation inhibition, or reduction of tRNA accessibility. mRNA degradation affects translation but is not the translation mechanism per se. While ribosome profiling may require extra efforts and not fit within the scope of the manuscript, the authors may consider it, or possibly testing the approach on a bi-cistronic system: RBS1-A-RBS2-B, and show whether targeting A only affects A. Of course, revising the text to tone down conclusiveness on translation control is also an option.

We thank the reviewers for their insights into potential mechanisms of SauCas9-mediated gene repression. In the absence of a PAM, we did not observe binding of SauCas9 to dsDNA in vitro. Thus, the activity of SauCas9 does not seem to be mediated by a CRISPRi-like, PAM-dependent inhibition of transcription. It is possible that SauCas9 blocks the ribosome (either at initiation or during elongation) or, as the reviewers suggest, by otherwise destabilizing the transcript. The text has been revised to include these possibilities and to emphasize the PAM-independent repression of gene expression, more broadly.